# Postharvest Flavor Quality Changes and Preservation Strategies for Peach Fruits: A Comprehensive Review

**DOI:** 10.3390/plants14091310

**Published:** 2025-04-26

**Authors:** Qiaoping Qin, Lili Wang, Qiankun Wang, Rongshang Wang, Chunxi Li, Yongjin Qiao, Hongru Liu

**Affiliations:** 1College of Ecological Technology and Engineering, Shanghai Institute of Technology, Shanghai 201418, China; qinqp@sit.edu.cn (Q.Q.); 19840154025@163.com (L.W.); wrs10123@163.com (R.W.); 2Crop Breeding & Cultivation Research Institute, Shanghai Academy of Agricultural Sciences, 1000 Jinqi Road, Fengxian District, Shanghai 201403, China; qiankunwang@saas.sh.cn; 3Institute of Shanghai Peach Research, NO.897, Jiangang Village, Laogang Town, Pudong New District, Shanghai 200120, China; hongruliu@saas.sh.cn

**Keywords:** peaches, preservation technology, aroma flavor, regulation, intelligent packaging

## Abstract

Peach (*Prunus persica* (L.) Batsch) is valued for its flavor, nutrition, and economic importance, yet as a climacteric fruit, it undergoes rapid postharvest senescence due to respiratory surges and ethylene production, leading to flavor loss and reduced marketability. Recent advances in postharvest physiology, including ethylene regulation, metabolic analysis, and advanced packaging, have improved preservation. Compared with traditional methods, emerging technologies, such as nanotechnology-based coatings and intelligent packaging systems, offer environmentally friendly and highly effective solutions but face high costs, technical barriers, and other constraints. This review examines changes in key flavor components—amino acids, phenolic compounds, sugars, organic acids, and volatile organic compounds (VOCs)—during ripening and senescence. It evaluates physical, chemical, and biotechnological preservation methods for maintaining quality. For instance, 1-MCP extends shelf life but may reduce aroma, underscoring the need for optimized protocols. Emerging trends, including biocontrol agents and smart packaging, provide a foundation for enhancing peach storage, transportation, and marketability.

## 1. Introduction

Peach (*Prunus persica* (L.) Batsch), a member of the Rosaceae family, is a deciduous fruit tree species originating from China with a cultivation history exceeding four millennia. Peach fruit is a rich source of essential nutrients and bioactive compounds, including organic acids, soluble sugars, vitamins (e.g., vitamin C), minerals (e.g., potassium), dietary fiber, and proteins. Notably, it contains significant amounts of polyphenols, flavonoids, and carotenoids, which exhibit potent antioxidant activities. These bioactive components contribute to multiple health benefits, such as promoting digestive function and enhancing immune response. Consequently, peach possesses substantial nutritional and economic value in the global fruit market. Commercially, peach fruit is categorized into five pomological groups: yellow-fleshed peaches, white-fleshed peaches, yellow-fleshed nectarines, white-fleshed nectarines, and percoche. While nectarines are botanically equivalent to peaches, they are distinguished by their smooth, trichome-free skin and exhibit nuanced differences in flavor profile and dimensional characteristics compared to their fuzzy counterparts. Percoche represents a specialized commercial variety characterized by firm, non-melting flesh texture, which is primarily utilized for fresh consumption and industrial processing applications [1].

Flavor profile serves as a crucial quality indicator for peach fruit, significantly influencing its market acceptability. The distinct flavor characteristics observed among different peach cultivars are primarily determined by the composition and concentration of key biochemical components, including amino acids, phenolic compounds, soluble sugars, organic acids, and volatile organic compounds (VOCs) [2]. As a typical climacteric fruit, peach undergoes significant physiological alterations postharvest, characterized by accelerated respiration rates and enhanced ethylene biosynthesis. These metabolic changes trigger rapid fruit senescence, resulting in substantial degradation of flavor compounds and quality deterioration [3]. About one-third of peach fruits are wasted during production and storage each year, ranging from 20% to 50% [4]. Consequently, developing effective strategies to modulate ripening processes during storage and maintain flavor stability and textural properties has emerged as a critical research focus in peach postharvest preservation. To address the challenges of postharvest softening and flavor degradation in peach fruits, researchers have developed various preservation strategies to extend storage life and enhance fruit quality. These approaches encompass physical, chemical, and biological methods, as well as their integrated applications, including low-temperature storage [5], Controlled Atmosphere storage [6], 1-methylcyclopropene (1-MCP) treatment [7], plant growth regulators (melatonin [MT] [8], methyl jasmonate [MeJA] [9], and salicylic acid [SA] [10]), and intelligent packaging systems [11]. These technologies primarily function through the regulation of ethylene biosynthesis and signal transduction, reduction in respiratory activity, and minimization of water loss, thereby effectively delaying ripening processes and maintaining nutritional and sensory quality.

In recent years, the consumption trend of peach fruits has shifted towards a significant upgrade in quality requirements. Consumers now prioritize high-sugar content, intense flavor profiles, attractive appearance, and good storage and transportation resistance when selecting peach varieties. Simultaneously, there has been a notable increase in attention towards green, organic, and standardized production methods. As the demand for high-quality agricultural products continues to rise, the preservation of flavor in horticultural products postharvest has become increasingly crucial [12]. However, current research indicates that preservation techniques differentially affect the flavor profiles of peaches. For instance, although cold storage, the most widely adopted preservation technique, can effectively suppress the respiratory metabolism and ethylene release of peaches, thereby inhibiting the ripening and senescence processes and extending the fruit’s shelf life, long-term cold storage may trigger chilling injury (CI) and lead to flavor deterioration in peach fruits [13]. Additionally, 1-MCP treatments have been found to potentially disrupt the synthesis and emission of volatile compounds, consequently altering the fruit’s sensory attributes [4]. These findings highlight the need for developing more precise and scientifically based preservation protocols that can effectively prolong storage duration while optimally maintaining the original flavor attributes of peach fruits.

This review systematically evaluates the impact of various preservation techniques on the flavor profiles, textural properties, and nutritional composition of peach fruits, based on the most recent scientific literature. The analysis aims to provide theoretical foundations and technical references for optimizing postharvest management strategies in the peach industry.

## 2. Progress in the Study of Postharvest Quality Changes in Peach

The quality attributes of peach fruit are predominantly determined by the dynamic composition of sugars, organic acids, amino acids, phenolic compounds (Table 1), and VOCs. During postharvest ripening, peaches undergo significant physiological changes, including accelerated respiratory metabolism, enhanced ethylene biosynthesis, texture softening, aroma volatile depletion, and alterations in sugar–acid balance. Understanding the transformation patterns of these key components during storage is crucial for optimizing storage parameters, extending shelf life, and maintaining optimal flavor quality.

This review systematically examines the postharvest changes in peach fruit quality through four key aspects: (1) amino acid metabolism, (2) phenolic compound dynamics, (3) sugar–acid balance, and (4) volatile aroma profiles. Furthermore, we will evaluate the regulatory effects of various preservation techniques on these quality parameters, providing comprehensive insights for postharvest management strategies.

### 2.1. Amino Acids

Amino acids serve as essential metabolic precursors for volatile compound biosynthesis, significantly contributing to the sensory attributes and nutritional value of horticultural products [20]. In plant systems, amino acids primarily exist in two distinct forms: protein-bound amino acids, which are incorporated into peptide chains or protein structures, and free amino acids (FAAs), which exist as unbound molecular entities within the cellular matrix [21]. FAAs serve dual functional roles in plant-derived foods: they not only function as readily bioavailable nutrients for human metabolism but also act as crucial taste-active components that significantly contribute to flavor profile development [15]. During ripening, amino acids serve as crucial precursors for the synthesis of aromatic compounds, which are further metabolized into higher alcohols, aldehydes, organic acids, phenols, and lactones [13]. Different types of FAAs contribute differently to the flavor of peach fruits, with glutamic acid and aspartic acid providing freshness; serine, glycine, threonine, and alanine providing sweetness; and leucine, isoleucine, arginine, valine, phenylalanine, and histidine providing bitterness [2,12]. Additionally, amino acid concentrations exhibit substantial variation among different peach varieties [22]. For instance, yellow-fleshed peaches generally contain higher total amino acid concentrations compared to other varieties, potentially resulting in a more pleasant aroma post-ripening [20]. Monti et al. [23] analyzed fifteen peach varieties and identified asparagine (Asn) as the most abundant amino acid, consistent with findings by Sun et al. [20]. Aspartic acid (Asp) and serine (Ser) were the next most prevalent. During postharvest ripening, alanine (Ala) levels increased in 13 varieties, serine (Ser) increased in 10 varieties, while proline (Pro) and glycine (Gly) decreased in 10 and 9 varieties, respectively. These findings highlight the dynamic changes in amino acid profiles during ripening and their potential impact on flavor development.

### 2.2. Phenolic Compounds

Phenolic compounds represent a diverse class of secondary metabolites synthesized through various plant metabolic pathways. These compounds exhibit multiple biological functions, including antioxidant activity, antimicrobial properties, pigmentation inhibition, and UV radiation protection. Furthermore, they play a significant role in shaping fruit sensory characteristics, particularly influencing astringency, bitterness, and aromatic profiles [24]. Phenolic compounds are mainly synthesized through the phenylpropanoid pathway, whereas the mevalonic acid and malonic acid pathways assume more indirect or supportive roles, depending on the specific biological context [25]. The fundamental chemical structure of phenolic compounds consists of an aromatic benzene ring with at least one hydroxyl substituent [26]. Phenolic compounds can be classified according to their structure into phenolic acids, astragals, coumarins, lignans, and flavonoids [24]. The main phenolic components of peach pulp are chlorogenic acid, catechins, epicatechins, and rutin, which account for about 70% of the total polyphenol content [16]. Optimal phenolic levels can enhance fruit flavor; however, excessive phenolic compounds—such as proanthocyanidins (soluble tannins)—may bind to salivary proteins, resulting in excessive astringency [27]. Comparative analyses have demonstrated that peach peel exhibits significantly higher concentrations of mineral elements, enhanced antioxidant capacity, and greater phenolic content compared to fruit pulp [28]. Also, there were significant differences in the phenolic content of different peach varieties, and all of them had strong antioxidant activity [29]. Zhao et al. [30] detected the presence of 495 phenolic metabolites in peach fruits, and the increase in anthocyanin synthesis in red-fleshed peaches synergistically promotes the accumulation of a variety of phenolic acids and flavonoids, which results in stronger antioxidant activity. It was found that fruit quality deterioration was accompanied by loss of phenolics, while phenolic content was closely related to the antioxidant capacity of the fruit [31]. Therefore, understanding the dynamic changes in phenolic compounds and their interaction with other components can help to analyze the quality of peach fruits.

### 2.3. Sugar–Acid Contents

The acidity and sweetness of the fruit are mainly determined by the sugar–acid contents, and the sugar–acid ratio is of great significance in evaluating the flavor and other sensory qualities of peach fruit. The major soluble sugars in peaches—sucrose, fructose, sorbitol, and glucose—exhibit dynamic changes in relative abundances during fruit maturation. In all varieties, glucose and fructose concentrations were nearly equivalent and lower than sucrose, whereas sorbitol levels were the lowest among these sugars [13]. Notably, immature fruits exhibit significantly higher proportions of glucose and fructose compared to their mature counterparts [18]. Peaches also contain low levels of organic acids (0.13–1.16%), mainly malic, quinic, and citric acids. Of these, malic acid is the most abundant and is the most important component influencing the acidic flavor of peach fruit [19]. Malic acid was significantly higher in the pulp than in the pericarp, whereas quinic acid, a precursor for the biosynthesis of phenolic compounds such as chlorogenic acid, was mainly present in the pericarp [32,33]. When compared with other fruit species such as kiwifruit and citrus, peaches exhibit relatively lower concentrations of L-ascorbic acid, despite its significance as a key antioxidant. Additionally, peaches contain modest levels of other organic acids including quinic acid, fumaric acid, and succinic acid [34,35]. Wu et al. [36] conducted a systematic analysis of seven peach cultivars in Xinjiang, China. The results revealed that fructose and sucrose account for over 70% of the total sugar content, while malic acid, quinic acid, and succinic acid constitute more than 80% of the total organic acids, playing a decisive role in determining the sweetness and acidity characteristics of peach fruits. It was found that the sugar fractions of peach fruits change with their ripening process, with glucose and fructose content rising as the fruit ripens, and sucrose content dominating when the fruit is fully ripe [18]. Wang et al. [37] found that the sucrose, glucose, fructose, and sorbitol contents of peach fruits stored at 12 °C versus 4 °C showed a decreasing trend. Zhou et al. [38] found that both the sucrose and sorbitol contents of peach fruits gradually decreased during cold storage, while fructose and glucose increased with the development of cold damage, suggesting that sucrose and sorbitol are converted to hexose, an energy metabolism substrate that maintains the energy balance of fruits to tolerate cold stress. Liu et al. [39] conducted an aroma omics study on peaches stored at 4 °C for 8 days after picking and found that during postharvest ripening, the content of sucrose, glucose, and fructose increased, while the content of sorbitol declined, contributing to the sweetness. Conversely, flavor deterioration occurred due to elevated isovaleric acid levels and amino acid metabolism. This study concluded that peach fruits exhibited optimal flavor when stored at 4 °C for 2–4 days after picking. These studies suggest that changes in the sugar–acid ratio of peach fruits during storage directly affect flavor perception, and how optimizing storage conditions to maintain the ideal sugar–acid ratio is key to future research.

### 2.4. Volatile Substances

VOCs play a pivotal role in determining fruit aroma quality. Postharvest physiological changes in peach fruits significantly alter the composition and concentration of volatile compounds, particularly affecting esters, alcohols, and aldehydes, which are the primary contributors to fruit aroma [40]. During postharvest storage, metabolic transformations of sugars, organic acids, and amino acids lead to the biosynthesis and release of diverse volatile compounds [41]. Over 100 volatile compounds contribute to peach aroma, with approximately 25 of these responsible for its characteristic fragrance [42]. Examples include alcohols, ketones, aldehydes, esters, lactones, carboxylic acids, phenols, and terpenes [40]. Fatty acids serve as critical precursors for certain volatile compounds in peach fruits—such as (*E*)-2-hexenol, γ-decalactone, (*Z*)-3-hexenal, and 2-hexenal—which are the primary contributors to the “fruity” or “grassy” aroma profiles of peaches [43]. Terpenoids and C13-norisoprenoids serve as critical “floral” aromatic components, with linalool prevalent in nectarines and β-ionone as well as its derivatives abundant in most peaches [43] (Table 2). Peach fruits of different varieties differ in the content of dominant volatiles. Studies have shown that red peach varieties contain higher levels of fruity volatiles—such as γ-decalactone and δ-decalactone—whereas white peach varieties exhibit greater abundances of grassy C6 compounds, including hexanal and trans-2-hexena [44]. Zhang et al. [1] concluded that the role of volatile compounds in flavor disparities between yellow- and white-fleshed peach varieties lies more in the abundance of specific volatiles than in the composition of unique ones. Studies have indicated that specific volatile compounds in peach cultivars are associated with fruit traits—such as firmness, color, and weight—though the underlying mechanisms require further elucidation [45].

Postharvest treatment technologies significantly influence the volatile composition of peach fruits. For instance, low-temperature storage has been shown to potentially suppress the biosynthesis of aromatic compounds, consequently affecting the overall flavor profile. Leng et al. [46] found a continuous increase in ester content and a gradual decrease in alcohols, aldehydes, and ketones in peach fruits during room temperature storage. Liu et al. [47] found that total VOC content in yellow peaches increased during ripening. Aldehydes dominated the volatile profile in early stages, accounting for 79–92% of total VOCs; by contrast, ester, and lactone levels rose to 53–60% and 25–38%, respectively, as fruits ripened, while aldehyde content declined to 2–12%. It has been found that alcohols can combine with acyl coenzyme A in cells to synthesize esters by alcohol acyltransferase, and it is thought that the decrease in alcohols may be associated with an increase in esters, particularly butyl acetate [39]. Therefore, an in-depth study of the dynamic changes in volatile substances and their relationship with other flavor substances is of great scientific significance for the postharvest preservation and quality assurance of peach fruits.
plants-14-01310-t002_Table 2Table 2Main aroma substances and their odor contributions.SpeciesRepresentativeOdorReferenceEsterHexyl acetateSpicy, banana, fruity aroma[39,48]Ethyl hexanoateFlower fragrance, greenness(Z)-3-Hexenyl acetateCandy fragranceEthyl caproateFruity aroma, sweetEthyl octanoateFruity, sweetLactoneγ-HexalactoneCoconut, vanilla, peach, sweet fragrance [48,49]δ-DecalactoneFruity, sweet[50]γ-DecalactonePeach-like, fruity, sweet[39]γ-OctalactoneCoconut, fruity[39]δ-OctalactoneCoconut, peach-like[39]Aldehyde2-HexenalGrassy, almondy[44,48,51]BenzaldehydeGrassy[48]HexanalGrassy[44,48]Terpeneβ-MyrceneResin, fruity fragrance[39,52]linaloolFlowery, fruity, woody[43]LimoneneOrange, lemon [39]


## 3. Advances in Freshness Preservation Technologies

### 3.1. Temperature-Controlled Preservation

Postharvest storage conditions significantly influence the flavor quality of peach fruits. Low-temperature storage remains the most prevalent preservation method due to its ability to inhibit microbial growth and reduce metabolic activity in horticultural products. This approach offers multiple advantages, including cost-effectiveness, operational simplicity, and preservation efficacy. However, as a chilling-sensitive fruit, peaches are prone to develop chilling injury symptoms at temperatures below 10 °C, with particularly severe damage occurring at 4–5 °C. Interestingly, the incidence of chilling injury can be mitigated when stored at 0 °C or near-freezing temperatures [53,54]. Cold damage to peach fruits manifests itself in the form of flesh browning, impaired ethylene synthesis capacity, reduced concentrations of total phenols, flavonoids, and total antioxidants, reduced lactones, esters, and terpenoids, and increased levels of aldehydes and alcohols, which ultimately leads to a significant loss of flavor [55,56], which can seriously affect the quality of peach fruits.

Cold shock (CS) treatment is a physical treatment that induces physiological resistance to storage physiological disorders by briefly exposing fruits and vegetables to ice water or cold air [57]. CS has been shown to be effective in attenuating cold damage, maintaining cell membrane integrity, and improving cold tolerance in peach fruit. By treating peach fruits with 0 °C ice water and 25 °C distilled water for 10 min, respectively, Ma et al. [58] showed that CS treatment effectively mitigated cold damage in peach fruits through multiple physiological mechanisms. This protective effect was achieved by increasing the activity of antioxidant enzymes, maintaining an optimal ratio of unsaturated to saturated fatty acids, and preserving cell membrane integrity. Jia et al. [59] immersed peaches in cold water at 0 °C for 10 min and stored them at 5.0 ± 1.0 °C for 35 days, demonstrating that CS treatment could reduce internal browning and improve cold tolerance of peach fruits during storage. Zhang et al. [60] found that cold water pre-cooling could reduce the respiration rate of yellow peaches, decrease water loss, and reduce the loss of ascorbic acid. Although low-temperature storage can effectively prevent the deterioration of peach fruit quality, choosing the appropriate storage temperature is a difficult problem to solve because the optimal storage temperatures of peach fruits of different varieties are different. Consequently, investigating appropriate low-temperature storage conditions and optimizing CS treatment parameters holds substantial promise for practical applications in the peach industry.

### 3.2. Gas-Conditioning Preservation Technology

Gas-conditioned storage maintains the flavor and nutritional composition of fruits and vegetables by regulating their environmental gas composition. Gas-conditioned storage according to the way to regulate the gas is divided into passive gas-conditioned preservation (Modified Atmosphere, MA) and active gas-conditioned preservation (Controlled Atmosphere, CA) categories. Passive air conditioning refers to the process by which fruits and vegetables consume oxygen in the packaging through their own respiration, generating carbon dioxide to create a low-oxygen, high-carbon dioxide environment; this inhibits the respiration of fruits and vegetables and extends shelf life. Active air conditioning, conversely, involves the artificial regulation of the storage atmosphere by injecting gases at specific proportions into the packaging (the most common are CO_2_, O_2_, and N_2_) [61]. Currently, gas-conditioning preservation technology has been widely used in the storage of perishable fruits such as peaches [62], whereby gas mixtures with tailored ratios are designed based on the specific physiological characteristics of fruits and vegetables to preserve their sensory qualities and nutritional integrity. Studies have shown that gas-conditioned storage can inhibit the oxidative decomposition of phenolic compounds in peaches [31], keep the content of volatile esters and lactones [62], and maintain the energy supply and sugar content of fruits [63]. Bulent et al. [64] found that MA treatment reduced the respiration rate and titratable acidity while maintaining higher sugar content, flesh firmness, vitamin C levels, and juice yield in the fruits. A. Veloso et al. [65] found that treating “Sweet Henry” peaches with a 2% O₂–15% CO₂ atmosphere during 56 days of storage enhanced and sustained fruit firmness, preserved acidity, and mitigated cold damage.

### 3.3. Packaging Technology

Packaging plays a pivotal role in the postharvest storage and marketing of fruits and vegetables. Due to their distinct gas-selective permeability properties, different packaging materials exhibit varying preservation efficiencies for horticultural commodities. When sealed in plastic films with suitable permeability, peaches leverage their respiratory activity to establish an optimal atmospheric composition within the package—one that balances oxygen and carbon dioxide levels. This process not only maintains internal humidity but also extends shelf life by regulating metabolic processes. Currently, commonly used film packaging materials include biaxially oriented polypropylene (BOPP), polyethylene (PE), polypropylene (PP), low-density polyethylene (LDPE), and polyvinyl alcohol (PVA), etc. However, traditional packaging materials are non-biodegradable, their recycling rate is low, and there is a big challenge to ecological environment protection, and thus degradable biopackaging technology will become the future development trend.

Active Packaging Materials (APMs) are an emerging food preservation technology designed to extend shelf life and maintain food quality by actively interacting with the food matrix. These materials can modulate the gas composition within the microenvironment of fruits and vegetables, release preservative agents, or absorb excess gases and moisture, thereby dynamically engaging with the food environment to enhance preservation efficacy. Electrostatic spinning is a major general-purpose technology used to design active packaging [66]. Cheng et al. [67] found that a PVA/CS/40% ICs nanofibrous film could extend the shelf life of strawberries up to 6 days at 25 °C.

As an emerging packaging material, coated films can act as a physical barrier against water and gas on fruit and vegetable surfaces, reducing water loss and oxidative damage. Additionally, these films can incorporate antimicrobial agents, antioxidants, and other food additives to maintain produce quality and extend shelf life [68]. Currently, common coating films include polysaccharides (pectin, starch, gum, alginate, chitosan, cellulose, etc.), proteins (gelatin, egg white, zein, whey proteins, casein, soy proteins, etc.), and lipid compounds (fatty acids, waxes, etc.), as well as composite coatings made from combinations of substances or materials [68]. Giuseppe et al. [69] found that aloe vera-based coatings, either alone or in combination with 1-MCP, effectively maintained the flavor quality of peach fruits. Jiao et al. [70] reported that a chitosan–chlorogenic acid graft copolymer (CS-g-CGA) coating effectively preserved postharvest firmness, soluble solids content, titratable acidity, and ascorbic acid levels in peach fruits. Some studies have shown that various nanoparticles (silver, gold, zinc, chitosan, platinum, iron, copper, and carbon nanotubes) exhibit synergistic antibacterial activity with essential oils (natural derivatives). An emerging technology involves using nanoparticles to encapsulate antimicrobial essential oils, whereby nanoparticle-based systems enhance the antibacterial and antifungal activities of these oils through various nanocomposite formulations [71]. Although film preservation is effective, its efficacy diminishes over time, and film safety requires strict auditing. Meanwhile, the strong aroma of essential oils may alter the intrinsic flavor of fruits and vegetables; thus, developing low-cost, long-acting, safe, non-toxic coated film preservatives with synergistic effects remains a critical challenge requiring breakthroughs. Intelligent packaging (IAP) is an advanced packaging technology with integrated sensors and monitoring devices to track, monitor, and manage packaged products [72]. Smart packaging comprises two primary types: information-responsive and intelligent controlled-release systems, which are applied in two distinct functional forms [11]. Information-responsive smart packaging utilizes indicator labels that react chemically with target compounds to generate color changes or other signals, thereby reflecting food freshness. The degree and nature of such color changes serve as a direct indicator of both food freshness and edibility safety (Figure 1). Intelligent controlled-release smart packaging functions by sensing environmental stimuli—such as temperature, humidity, or gas composition—to release active agents, thereby mitigating adverse effects of environmental stresses on food quality [72]. For example, Wang et al. [73] developed a novel chitosan-based packaging film with exogenously triggered controlled-release functionality by integrating a cinnamaldehyde nanocapsule (SNC) emulsion into a chitosan (CS) membrane matrix, thereby providing a new controlled-release strategy for antimicrobial agent delivery. Intelligent packaging technology, by integrating sensors, smart labels, wireless communication, and other advanced technologies, continuously monitors and records environmental parameters—such as temperature, humidity, and gas composition—during fruit and vegetable storage. This technology can even dynamically regulate internal package conditions, including adjusting gas mixtures and releasing antimicrobial agents, showcasing substantial promise for applications in produce preservation [74].

### 3.4. Exogenous Hormone Treatment

Exogenous hormone treatment, as one of the most widely used food preservation techniques, effectively maintains the quality of fruits and vegetables, reduces nutrient loss, and extends storage duration. Commonly applied exogenous hormones include ethylene, 1-methylcyclopropene (1-MCP), melatonin (MT), and methyl jasmonate (MeJA).

As a competitive inhibitor of ethylene signaling, 1-MCP has been widely used in the postharvest preservation of horticultural commodities [7]. Wang et al. [2] showed that 1-MCP treatment increased levels of desirable flavor compounds such as linalyl acetate and sucrose while decreasing undesirable ones, including benzaldehyde and histidine. Cai et al. [75] found that 1-MCP inhibited organic acid degradation in peach fruits by delaying the expression of genes encoding malate- and citrate-degrading enzymes, thereby maintaining higher organic acid content. This aligns with Zhou et al. [76], who demonstrated that 1-MCP treatment delayed soluble sugar metabolism by upregulating genes involved in glucose and sorbitol biosynthesis, thereby maintaining malic and citric acid concentrations in fruits during the late storage period. The treatment also inhibited organic acid degradation, preserving the flavor quality of peach fruits. However, some studies have reported that 1-MCP treatment inhibits the synthesis of peach-specific aroma, especially volatile esters [77], and retains higher levels of green aroma [78]. This is hypothesized to occur because 1-MCP delays ethylene peak formation and reduces ethylene production by downregulating genes involved in volatile biosynthesis and ethylene signaling [79], thereby disrupting the metabolic pathways that generate characteristic peach aroma compounds. 1-MCP also inhibits the release of aromatic volatiles in both intact and fresh-cut peaches, compromising overall organoleptic quality, and exacerbating pulp browning [80]. Thus, optimizing the 1-MCP application to preserve peach fruit quality while maintaining desirable aroma profiles warrants further investigation.

MT, a crucial phytohormone regulator, plays a significant role in modulating auxin metabolism and enhancing plant stress tolerance [81]. As an essential signaling molecule, MT strengthens both enzymatic and non-enzymatic antioxidant systems, thereby facilitating the scavenging of reactive oxygen species (ROS) during fruit storage [82]. Cao et al. [83] applied MT treatment to postharvest peach fruits and demonstrated that MT upregulated genes involved in antioxidant pathways, thereby alleviating cold-induced damage. Wu et al. [84] found that MT treatment increased the content of total soluble protein and glutamate in yellow-fleshed peaches, reduced total free amino acid levels, enhanced fruit antioxidant activity, and preserved fruit quality and nutritional value. Bao et al. [85] found that melatonin-treated peach fruits exhibited significantly higher total phenolic content and reduced cold damage incidence.

Methyl jasmonate (MeJA), a naturally occurring phytohormone that is exogenously applied to fruits, effectively enhances fruit quality. Studies have demonstrated that MeJA-mediated enhancement of cold hardiness in peach fruits involves regulatory mechanisms linked to ethylene and sugar metabolism [86]. Cai et al. [9] reported that MeJA treatment effectively reduced cold damage incidence and regulated ethylene metabolism while inducing the biosynthesis of aromatic lactones in peach fruits during cold storage. These results align with Duan et al. [87], who demonstrated that MeJA-treated peaches exhibited significantly higher levels of fruity esters and lactones than controls, thereby minimizing the loss of aroma-related volatile compounds.

### 3.5. Combined Treatment

Postharvest preservation and quality deterioration of peach fruits are influenced by multiple factors, and each preservation technique has inherent limitations. Consequently, considerable attention has focused on developing integrated preservation strategies that integrate multiple technologies to maximize synergies, thereby enhancing the effectiveness of extending the postharvest shelf life of peach fruits. Huang et al. [88] found that 1-MCP fumigation plus nano-material packaging (1-MCP-NA) was able to effectively delay and inhibit the reduction in esters and aldehydes in postharvest peach fruits. Li et al. [89] reported that the combined application of 1-MCP and UV-B treatments on fresh-cut peaches suppressed phenolic metabolism and retarded respiratory activity. Zhao et al. [90] evaluated the efficacy of hot-air treatments, antagonistic yeast, and their combination against two major postharvest diseases of peach fruits. They found that the combined treatment of hot air and Pichia guilliermondii significantly inhibited wound infections in peach fruits, demonstrating greater efficacy than either treatment alone. Jiao et al. [70] treated postharvest peach fruits with a chitosan-grafted chlorogenic acid complex (CS-G-CGA) and stored them at 20 °C for 8 days. This study found that fruits treated with CS-G-CGA exhibited significantly higher antioxidant activity than those treated with chitosan alone, along with better retention of fruit firmness, soluble solids content, titratable acidity, and L-ascorbic acid levels. Wu et al. [91] reported that the combined treatment of 1-MCP and NO better maintained fruit firmness, antioxidant enzyme activity, and lower ROS content compared with individual treatments, demonstrating superior preservation efficacy.

Diverse postharvest preservation technologies display unique advantages and limitations in extending the shelf life and preserving the flavor quality of peach fruits. The selection of suitable techniques or integrated strategies must be tailored to specific fruit attributes, market demands, and operational parameters. Table 3 and Figure 2 provide a comprehensive summary of the commonly used preservation methods discussed in this review, along with their respective advantages and limitations. Figure 3 represents the effect of different storage techniques on peach fruit quality.

## 4. Summary and Prospects

This review offers a thorough exploration of recent progress in understanding postharvest quality changes and preservation technologies for peach fruits. It provides an in-depth analysis of the dynamic changes in amino acids, phenolic compounds, sugar–acid ratios, and VOCs during postharvest storage—key factors that collectively influence the flavor profile and commercial value of peaches. The article systematically assesses the impacts of diverse preservation strategies on peach quality, including physical techniques (such as temperature regulation and Modified Atmosphere storage), chemical treatments (like 1-MCP treatments), and biotechnological innovations (such as active packaging and edible coatings). Each method presents unique strengths and limitations and their combined use plays a pivotal role in extending shelf life, maintaining quality attributes, and retarding physiological deterioration.

Despite notable progress in enhancing storage stability through current preservation technologies, future research is increasingly focused on developing smart packaging systems and integrating molecular biology techniques to optimize storage efficiency. Key priorities include the creation of eco-friendly preservatives and a deeper understanding of consumer preferences to align with market demands. Cutting-edge innovations, such as nanotechnology-based films and coatings, offer precise control over gas exchange and microbial growth at the molecular level, thereby preserving fruit freshness. Nanotechnology-based active and smart packaging systems have diverse applications in food packaging, including (1) using nanoparticles as carriers for bioactive materials; (2) incorporating active nanoparticles into polymer matrices to enhance packaging performance; and (3) developing smart packaging equipped with nanosensors for real-time quality monitoring. These advancements have already yielded substantial results across various food preservation and processing sectors, providing a robust foundation for future breakthroughs. Additionally, the integration of machine learning algorithms to streamline harvest, storage, transportation, and marketing processes—paired with big data analytics for quality forecasting and real-time supply chain optimization—holds enormous potential for minimizing postharvest losses.

By leveraging advanced technologies and novel approaches, this field is poised to deepen scientific understanding of peach preservation while addressing practical challenges directly. Such initiatives are instrumental in enhancing the market competitiveness of peaches and related fruit products, meeting consumers’ growing expectations for superior quality. Looking forward, research should prioritize the convergence of state-of-the-art technological advancements to drive innovation in peach preservation, ultimately fostering the sustainable growth of the fruit industry.

## Figures and Tables

**Figure 1 plants-14-01310-f001:**
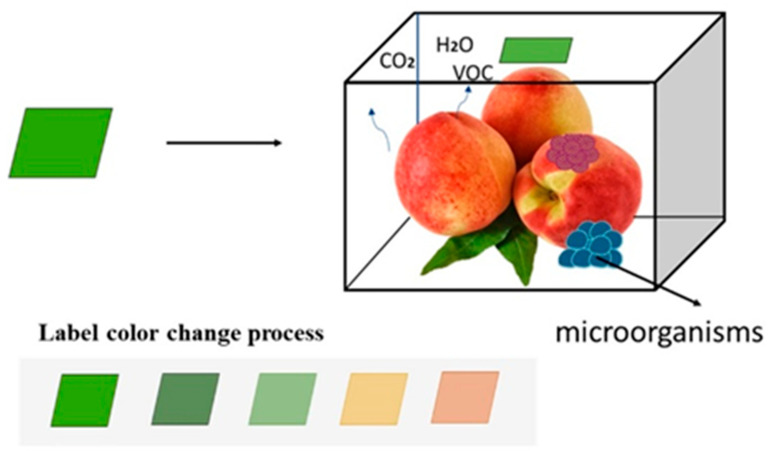
Example of the application of a carbon dioxide indicator in peach preservation.

**Figure 2 plants-14-01310-f002:**
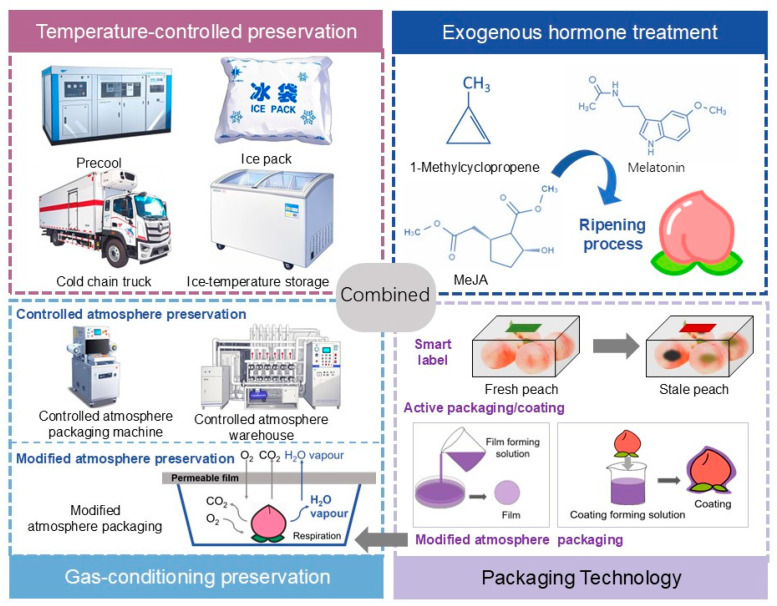
Summary of different storage technologies.

**Figure 3 plants-14-01310-f003:**
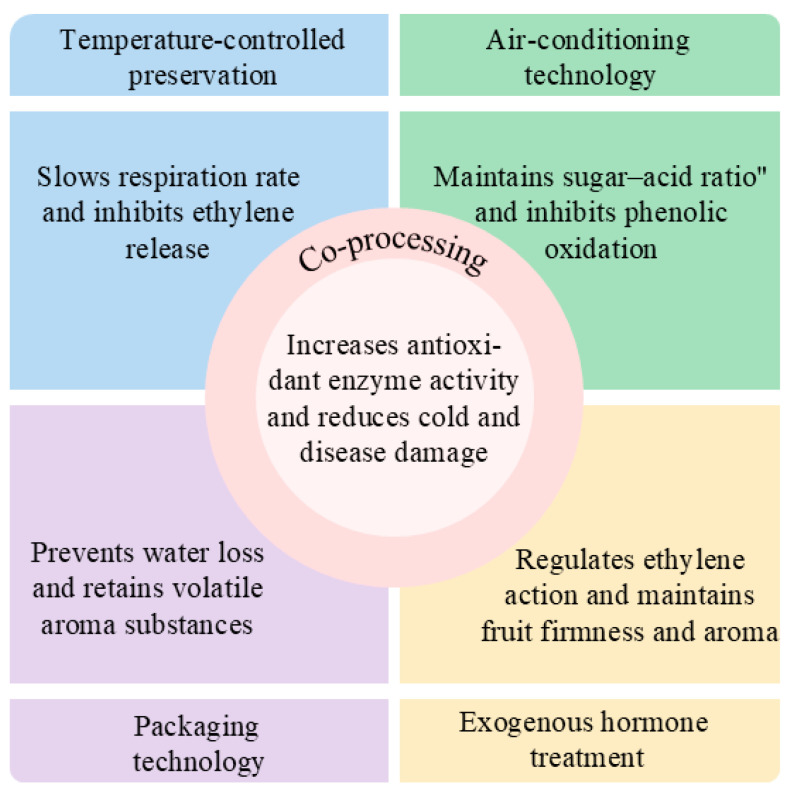
Effect of different storage techniques on peach fruit quality.

**Table 1 plants-14-01310-t001:** Main representative components of peach non-volatile flavor substances and their flavor effects.

Peach Non-Volatile Flavor Components	Main Representative Substances	Affects Flavor	Species Examples	Reference
Amino acids	Pro	Sweet	Golden Honey Peach	[12,13,14,15]
	Glu	Umami	Yangshan Honey Peach	
	Asp	Sweet, sour	Feicheng Peach	
	Asn	Bitter, astringent	Yellow-fleshed Peach	
	Ala	Pure and sweet	Yulu Flat Peach	
	Arg	Slightly bitter	Qingzhou Green Peach	
Phenolics	Chlorogenic acid	Slightly bitter	Jjubao Peach	[16,17]
	Neochlorogenic acid	Bitter	Golden Peach	
	Epicatechins	Acerbic	Red-fleshed Peach	
	Anthocyanins	Acerbic	Red-fleshed Peach, Yellow-fleshed Peach	
	RutinProanthocyanidins	Acerbic	Blood peach	
Sugars	Sucrose	Sweet	Jinxia	[13,18]
	Glucose	Sweet	Yixianghong	
	Fructose	Sweet	Touxinhong	
	Sorbitol	Sweet	Yixianghong	
Acids	Quinic acid	Sour	Yixianghong	[13,19]
	Malic acid	Sour	NJN76	
	Citric acid	Sour, acerbic	Tropic Prince	

**Table 3 plants-14-01310-t003:** Postharvest preservation techniques for peach fruits.

Preservation Technology	Main Methods	Dominance	Limitations	Reference
Temperature-controlled preservation	Low-temperature storage, near-freezing temperature, CS, heat treatment	Low cost, easy to operate, widely applicable	Low temperatures are prone to cold damage and heat treatment may affect flavor	[57,58]
Air-conditioning technology	MA, CA	Effectively extended shelf life, maintained nutrition and flavor	High equipment costs and precise control of gas ratios	[61]
Packaging technology	Active packaging, smart packaging, coating technology	Extended shelf life and increased market value	Certain materials may affect the natural flavor and cost more	[11,74,92]
Exogenous hormone treatment	Ethylene/1-MCP, MeJA, MT treatment	Regulation of the postharvest ripening process, specifically targeted	Certain hormones may inhibit aroma release and affect overall flavor	[7,80,85]
Combined treatment	Combined application of multiple technologies	Combined effect and extended shelf life	Requires optimized combination of solutions and is complex to operate	[87,88]

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
