# Peer review of "Postharvest Flavor Quality Changes and Preservation Strategies for Peach Fruits: A Comprehensive Review"

_plants, 2025, doi:10.3390/plants14091310_

Round 1

Reviewer 1 Report

Comments and Suggestions for Authors

The article “Postharvest Flavor Quality Changes and Preservation Strategies for Peach Fruits: A Comprehensive Review” provides state-of-the-art progress and reviews the progress made in understanding the postharvest physiology of peach fruit, preservation technology upgrades, innovative emerging trends in preservation, and technologies to maintain the quality of Prunus persica (L.) Batsch.

The topic is timely and provides significant details on maintaining the postharvest quality of peaches and addresses existing bottlenecks in preservation, an important factor in postharvest quality in fruits and vegetables.

I have some suggestions and queries for the improvement of the manuscript.

Abstract section: Compare and contrast the traditional preservation methods with advanced biotechnological approaches. In addition, current progress/achievements and future directions need to be discussed precisely.

The first mention of peach should be in scientific form, then an abbreviated form may be used.

In the introduction section, very few reference studies were cited. There are no references. What is the basis of the introductory literature and findings?

Line 48-50, Consequently, developing effective strategies……….no reference?

Line 5358: These approaches encompass physical, chemical, and biological methods. Please provide the reference study and likewise for key information provided in the article.

Line 63-64: the current research indicates preservation techniques differentially affect the flavor profiles, how? By far which is the most successful preservation method for peaches and the commercial varieties? Discuss.

The literature study focuses on peach fruit. It is important to provide names and discuss the key fruit varieties of commercial interests, marketed globally in the plant overview. In addition, what are the marketing trends?

What are the key outcomes in the development/advanced technologies for postharvest fruit quality maintenance? The conclusion should prioritize the actionable goals, e.g., key progress in nanotechnology-based coatings and films in food preservation, and advanced machine learning algorithms (case studies) should be discussed as an example.

Table 3. has no cited references and needs to be revised to include relevant studies.

Moderate English revisions are required to improve clarity.

For e.g. line 163-164: stored at 12°C versus 4°C………….,  while in line 165: peaches stored at 4 degrees Celsius………….., please be consistent in the use of abbreviated forms.

Table 2. Needs to be revised for clarity. The aromatic constituents should be classified. Row 1 and row 2- the literature is mixed (a 3-line table orientation can be used). Besides, it is also important to add the fruit varieties and then differentiate their aromatic constituents for better presentation.

All the figures/diagrammatic representations are well presented and discuss key pieces of information.

References: Author names and DOIs are missing, and incomplete references. Please revise as per MDPI guidelines.

Comments on the Quality of English Language

Moderate English revisions are necessary for improved clarity of the manuscript.

Reviewer 2 Report

Comments and Suggestions for Authors

This review provides a thorough and well-structured overview of postharvest flavor changes and preservation strategies in peach fruits. The manuscript is scientifically sound, well-referenced, and addresses an important topic in postharvest biology. However, some sections could benefit from additional depth, clarity, or reorganization to enhance readability and impact. Below are specific suggestions for improvement.

1.The introduction section should expand on the global economic impact of peach production and postharvest losses to underscore the urgency of the topic. Add briefly mention consumer trends to justify the focus on advanced preservation technologies.

  1. 2.1. Amino Acids section should add a comparative table showing FAA profiles across peach cultivars.
  2. Discuss practical implications of phenolic degradation, please.
  3. Clarify why sorbitol declines during storage.
  4. Compare CS treatment protocols, e.g., immersion time, temperature gradients.
  5. Improve Figure 1 readability.
  6. Define acronyms at first use.
  7. Update citations to include 2024–2025 studies on nanotechnology/VOCs.

Round 2

Reviewer 1 Report

Comments and Suggestions for Authors

Thank you for your efforts in the precise and point-by-point revision of the manuscript. It can be accepted for publication in the present form.

Reviewer 2 Report

Comments and Suggestions for Authors

The author has revised the review comments very well. It is recommended to accept them.